# MLAM: Multi-Layer Attention Module for Radar Extrapolation Based on Spatiotemporal Sequence Neural Network

**DOI:** 10.3390/s23198065

**Published:** 2023-09-25

**Authors:** Shengchun Wang, Tianyang Wang, Sihong Wang, Zixiong Fang, Jingui Huang, Zuxi Zhou

**Affiliations:** 1College of Information Science and Engineering, Hunan Normal University, Changsha 410081, China; scwang@hunnu.edu.cn (S.W.); 202170293855@hunnu.edu.cn (T.W.); 202120293802@hunnu.edu.cn (S.W.); td20210006@hunnu.edu.cn (Z.F.); 2College of Civil Aviation, Nanjing University of Aeronautics and Astronautics, Nanjing 211106, China; zhouzuxi@126.com

**Keywords:** precipitation nowcasting, radar echo extrapolation, convolutional recurrent neural networks, attention mechanism

## Abstract

Precipitation nowcasting is mainly achieved by the radar echo extrapolation method. Due to the timing characteristics of radar echo extrapolation, convolutional recurrent neural networks (ConvRNNs) have been used to solve the task. Most ConvRNNs have been proven to perform far better than traditional optical flow methods, but they still have fatal problems. These models lack differentiation in the prediction of echoes of different intensities, which leads to the omission of responses from regions with high intensities. Moreover, because it is difficult for these models to capture long-term feature dependencies among multiple echo maps, the extrapolation effect declines sharply over time. This paper proposes an embedded multi-layer attention module (MLAM) to address the shortcomings of ConvRNNs. Specifically, an MLAM mainly enhances attention to critical regions in echo images and the processing of long-term spatiotemporal features through the interaction between input and memory features in the current moment. Comprehensive experiments were conducted on the radar dataset HKO-7 provided by the Hong Kong Observatory and the radar dataset HMB provided by the Hunan Meteorological Bureau. Experiments show that ConvRNNs embedded with MLAMs achieve more advanced results than standard ConvRNNs.

## 1. Introduction

Precipitation nowcasting refers to the prediction of precipitation in a certain area in the next 2 h using available meteorological data. Severe convective weather, such as heavy precipitation, is characterized by rapid development and strong destructiveness, which are shown in precipitation nowcasting. The accuracy, reliability and timeliness of precipitation nowcasting are essential to mitigate the impact of severe convective weather events on the daily lives of many. At present, precipitation nowcasting is mainly based on radar echo extrapolation, that is, using the radar echo images generated by weather radars to predict the echo distribution in the future [1]. Specifically, according to the characteristics of the echo of precipitation particles, weather radars measure the reflectivity of precipitation particles by emitting electromagnetic waves and receiving echoes, thereby generating radar echo intensity maps. The echo intensity distribution reflects current weather conditions [2]. In general, more precipitation occurs in areas with greater radar echo intensity.

Radar echo extrapolation methods are mainly divided into traditional methods and extrapolation methods based on deep learning. The traditional radar echo extrapolation algorithms mainly include the cross-correlation method, cell centroid method and optical flow method [3,4,5,6,7]. However, there are three obvious limitations to traditional radar extrapolation methods. Firstly, these traditional methods extrapolate from adjacent radar echo images and cannot accurately capture the motion state of echoes, such as maintenance, development and dissipation. Second, echoes located at the edge of the weather radar observation area are easily ignored. Third, they cannot take full advantage of the large number of radar echo images, and the huge amount of computation adversely affects the timeliness of extrapolation.

With the improvement in big data processing capabilities, deep learning has developed rapidly and been applied to a variety of tasks to improve performance, such as in image processing, machine translation, intelligent healthcare and other fields. The deep learning model is good at learning the basic characteristics and change trends of data collectives from massive data, and its development provides a new method for studying radar echo extrapolation [8]. The radar echo extrapolation algorithm based on deep learning studies the evolutionary development characteristics of echoes from large-scale radar echo sequence data and then automatically predicts future echo images. Because the development of radar echoes has a causal relationship with time, Shi et al. [9] defines the radar echo extrapolation task as a spatiotemporal sequence prediction problem in the field of deep learning. Shi et al. [9] proposed a convolutional long short-term memory network (ConvLSTM), which uses convolutional structures to replace the fully connected structures in a long short-term memory network (LSTM) [10] to process the echo images, emphasizing the importance of convolution in capturing spatial changes. The extrapolation effect of ConvLSTM is significantly stronger than that of the optical flow-based extrapolation method and fully connected LSTM method, which indicates that the deep learning model has great potential in handling the radar extrapolation task. Currently, most radar extrapolation methods based on deep learning are mainly implemented based on convolutional neural networks (CNNs) and recurrent neural networks (RNNs).

The radar extrapolation method based on CNNs can capture the spatial features of the echo pattern well, but it is difficult to deal with the time features contained in the echo sequence. Agrawal et al. [11] believes that echo extrapolation is an image–image conversion problem and used U-Net [12] to perform short-term echo extrapolation, which is superior to traditional extrapolation methods in various indicators. Song et al. [13] and Kevin et al. [14] used U-Net as the core network and added Squeeze-and-Excitation Networks (SENets) [15] and a convolutional block attention module (CBAM) [16] to it, respectively, which greatly reduced the parameter size of the original U-Net and improved the prediction effect. Most subsequent studies combine various functional modules, aiming to trade complex network structures for better predictions. The SimVP [17] designed by Gao et al. consists of an encoder, a translator and a decoder. The encoder is used to extract spatial features, the translator learns time evolution and the decoder integrates spatiotemporal information to predict future frames. This algorithm uses the simplest CNN structure to achieve a relatively advanced prediction level.

The radar echo extrapolation method based on RNNs has greater advantages for processing echo sequences. Most existing spatiotemporal sequence networks are improved based on ConvLSTM [9]. PredRNN [18] introduces a new spatiotemporal memory state, which increases the vertical information propagation path in the network and enhances the exchange of information among layers. PredRNN++ [19] adjusts the spatiotemporal memory state and the temporal memory state to cascade, thereby increasing the propagation depth of the network. At the same time, the Gradient Highway Unit (GHU) is embedded between the lowest layers of the network to provide a fast track for backpropagation. In order to utilize the differential signals of adjacent moments, MIM [20] adds two self-updating memory modules to process the high-order nonlinear spatiotemporal features in the sequence. Trajectory GRU (TrajGRU) [21] applies learnable convolution to capture the spatiotemporal correlation among distant features and improves the traditional Encoding-Forecasting structure. EIDETIC 3D LSTM (E3D-LSTM) [22] utilizes 3D convolution and a self-attention gated unit to process the memory information of historical moments, which improves the ability of the network to perceive local motion. Sun et al. [23] took radar echo images of various heights as the research object, first extracting spatial features of the echoes by 3D convolution and then predicting echoes by 3D-ConvLSTM. MotionRNN [24] emphasizes the impact of instantaneous changes in spatiotemporal sequences on prediction, so the new MotionGRU captures complex changes and can uniformly model transient changes and movement trends. For radar echo extrapolation tasks, the visualization of extrapolation results often cannot meet the requirements of meteorological observation. Xu et al. [25] integrated a Generative Adversarial Network (GAN) into LSTM and enhanced the quality of the predicted image using a fixed parameter generator. Ravuri et al. [26] proposed deep generative models of radar (DGMRs), which focus on the prediction of rare moderate-to-heavy rainfall and show advantages in short-term prediction tasks. Effective handling of long-term global dependencies is the key to obtaining better prediction results. The development of various attention mechanisms contributes to the improvement of spatiotemporal sequence prediction tasks. Self-Attention ConvLSTM (SA-ConvLSTM) [27], interactional dual attention long short-term memory (IDA-LSTM) [28], spatiotemporal convolutional long short-term memory ST-ConvLSTM [29] and TA-ConvLSTM [30] extend ConvLSTM by integrating the attention mechanism inside the network unit. SA-ConvLSTM applies a self-attention module (SAM) [31] to memorize features that have long-term dependencies in spatial and temporal domains. Similarly, Luo et al. deployed the dual attention mechanism to enhance the focus on high-intensity echoes. ST-ConvLSTM can effectively perceive local and global spatiotemporal changes. TA-ConvLSTM adds a time-dependent capture module and attention mechanism, which improves the network’s ability to capture changing features of radar echoes. The attention mechanism in the above model is mainly integrated within the unit, which limits the scope of processing information in the network and is difficult to adapt to other spatiotemporal sequence network units.

However, as shown in Figure 1, there are two common problems with these previous studies of echo extrapolation using spatiotemporal sequence networks. First, the spatiotemporal sequence network is stacked in multiple layers, and it is difficult for low-level information to pass to the output layer. Therefore, it is difficult to make full use of the original information. Specifically, radar echo intensity shows a rapid decline with extrapolation. Second, critical echo locations cannot be adequately addressed during extrapolation, such as high-intensity echo regions (meaning areas of heavy precipitation) and edge areas where echo intensity changes. Therefore, more and more detailed information is lost in the forecast, and echo images are gradually blurred until they are meaningless.

In this paper, we propose a multi-layer attention module (MLAM). MLAMs not only alleviate the loss of important features faced by spatiotemporal sequence networks but can also be deployed as a general module in multiple spatiotemporal sequence networks. The main contributions of this paper are as follows:We propose an improved cascaded dual attention module (CDAM) based on a self-attention module (SAM). CDAM can automatically adjust the feature weights of different positions in radar echo images and enhance the causal relationship between different features. Compared with other attention mechanisms, CDAM has a stronger ability to extract and abstract features. Therefore, it can enhance important details and slow down the blurring of extrapolated images.We combine the 3D convolutional block attention mechanism (3D-CBAM) and CDAM to propose a versatile MLAM. It strengthens the processing of key echo regions from a global perspective and grasps the trend of echoes.We design a new architecture so that the MLAM can be applied to ConvRNNs. Contrast experiments were conducted between ConvRNNs and ConvRNNs embedded with MLAMs. ConvRNNs with MLAMs achieve better results than standard ConvRNNs on radar echo datasets provided by the Hunan Meteorological Bureau and Hong Kong Observatory.

## 2. Background and Data

### 2.1. Background

#### 2.1.1. ConvRNNs

Figure 2 illustrates that the basic model is a multi-layer network stacked by ConvRNNs. The basic building block of ConvRNNs is the current mature spatiotemporal sequence network units, such as ConvLSTM and ST-LSTM.

Specifically, the input and output of ConvRNNs are sequence data. In the vertical direction, information is propagated from the bottom up. The bottom layer processes the original input information, and the output of the top layer is the prediction of the next moment. By stacking layer by layer, ConvRNNs can fully extract important spatial feature information at multiple scales in each frame. In the horizontal direction, in order to satisfy the characteristics of long-term memory of spatiotemporal sequence tasks, the memory unit is used to record the important features of historical information.

ConvLSTM is used as an example to illustrate the information transmission and update the formula of the basic model. The key equations of ConvLSTM are shown as follows:(1)$$i_{t}=\sigma (W_{xi}*X_{t}+W_{hi}*H_{t-1}+W_{ci}⨀C_{t-1}+b_{i})$$
(2)$$f_{t}=\sigma (W_{xf}*X_{t}+W_{hf}*H_{t-1}+W_{cf}⨀C_{t-1}+b_{f})$$
(3)$$C_{t}=f_{t}⨀C_{t-1}+i_{t}⨀\tanh(W_{xg}*X_{t}+W_{hg}*H_{t-1}+W_{cg}⨀C_{t-1}+b_{g})$$
(4)$$o_{t}=\sigma (W_{xo}*X_{t}+W_{ho}*H_{t-1}+W_{co}⨀C_{t-1}+b_{o})$$
(5)$$H_{t}=o_{t}⨀\tanh(C_{t})$$

Here, the input $X_{t}$, the memory states $C_{t}$ and the hidden states $H_{t}$ of the timestamp t are all 3D tensors. As gated units, $i_{t}$,$f_{t}$ and $o_{t}$ play different roles in the dissemination of information. ∗ and ⨀ denote the convolution operator and the Hadamard product.

#### 2.1.2. Self-Attention Module

Each feature processed by the self-attention module (SAM) records a correlation with the features of all locations as shown in Figure 3, which helps to handle long-distance dependencies in space. From the perspective of the image domain, each pixel in the image establishes a relationship with other pixels. It helps to expand the receptive field of features and obtains more contextual information during image processing. The input feature $H_{t}\in R^{c\times N}$ is mapped into three feature maps with different effects through 1×1 convolution, namely $V_{h}\in R^{c\times N}$, $K_{h}\in R^{c\times N}$ and $Q_{h}\in R^{c\times N}$, where c represents the channel and N=h×w. The similarity score matrix $S_{h}\in R^{N\times N}$ is obtained by calculating the similarity for each pixel through the matrix product.
(6)$$S_{h}=Q_{h}^{T}K_{h}.$$

The similarity score of each position is normalized by using the Softmax function and $S_{h}^{\prime }$ $\in R^{N\times N}$ is the normalized matrix,
(7)$$S_{h}^{\prime }=Softmax(S_{h}).$$

The feature of each location $A_{h}\in R^{c\times N}$ is determined by the weighted sum of all locations,
(8)$$A_{h}=S_{h}^{\prime }V_{h}.$$

In order to ensure the stability and flexibility of the module, the final output ${\hat{H}}_{t}\in R^{c\times N}$ is obtained by the residual connection between the input features $H_{t}$ and $A_{h}$,
(9)$${\hat{H}}_{t}=H+A_{h}.$$

#### 2.1.3. The 3D-Convolutional Block Attention Module

The convolutional block attention module (CBAM) is a simple and efficient fusion attention module that can be directly applied to various networks. It is composed of the channel attention module and the spatial attention module and strengthens important features from the two dimensions of channel and space. Different from CBAM, as shown in Figure 4, 3D-CBAM can extract not only spatial features but also feature relations between frames [32].

As shown in Figure 5, the input feature map firstly performs global average pooling and global maximum pooling in parallel. Then, a multi-layer perceptron (MLP) is used to reduce the number of parameters of the two feature maps. A new feature map is calculated by adding the corresponding values of the two feature maps. Then, it passes through the sigmoid function to the channel attention feature map $M_{c}$. The channel attention feature map is used to adjust the weight distribution of each channel, which makes the model pay more attention to important features. Finally, the final output of the channel attention module is obtained by multiplying the channel attention feature map with the original input feature.
(10)$$M_{c}=\sigma (\mathrm{MLP}(MaxPool3D(F))+\mathrm{MLP}(AvgPool3D(F))).$$
(11)$$F\prime =M_{c}⨂F.$$

The spatial attention module focuses on pixel positions that are decisive for the prediction in Figure 6. The output of the channel attention module (CAM) serves as the input of the spatial attention module (SAM). After entering the SAM, the feature map is first implemented with maximum pooling and average pooling based on the channel dimension. Then, two feature maps obtained by pooling are stacked along the channel direction. The stacked feature map is reduced to a single channel by the convolutional layer. The spatial attention map is generated after the sigmoid function. Finally, the spatial attention feature map is multiplied with the input of the module to obtain the final feature map.
(12)$$M_{s}=\sigma (Conv([MaxPool3D(F\prime ),AvgPool3D(F\prime )]).$$
(13)$$F\prime \prime =M_{s}⨂F\prime .$$

### 2.2. Data

Two radar echo datasets are used in this paper, which are the HKO-7 provided by the Hong Kong Observatory and the HMB provided by the Hunan Meteorological Bureau. The following reasons show why we chose the HKO-7. Firstly, Shi et al. defined the precipitation prediction as a spatiotemporal sequence prediction task for the first time and validated ConvLSTM by using part of the data from HKO-7 [14]. Later, the HKO-7 dataset was presented as a benchmark dataset to standardize subsequent studies [21]. Therefore, HKO-7 is representative in the field of rainfall prediction. Secondly, most of the comparison models selected in this paper are evaluated on the HKO-7 dataset. HKO-7 can effectively detect the performance of an MLAM and enhance the comparability of results. The reason for choosing HMB is that it can provide rich and varied radar echo samples because of the abundant precipitation in Changsha, Hunan Province. More importantly, this study aims to improve the accuracy of heavy precipitation nowcasting in Hunan, so the proposed method should be more suitable for the precipitation situation in Hunan.

#### 2.2.1. Data Sources

HKO-7: HKO-7 was collected by the Hong Kong Observatory using the S-band single-polarization weather radar, which includes radar echo images from 2009 to 2015. The radar CAPPI images have an altitude of 2 km and cover an area of 512 km × 512 km in Hong Kong. A frame of radar echoes is obtained every six minutes, with a total of 240 frames throughout the day, each with a resolution of 480 × 480 pixels [21].

HMB: The HMB was collected by several S-band weather radars provided by the Hunan Meteorological Bureau located in Changsha, Hunan Province (105.00°~117.09° E, 21.99°~33.01° N). The dataset spans three years (2018–2020), which ensures the diversity of radar echo images. Moreover, there is a high temporal resolution of 6 min and a high spatial resolution of 2.5 km in the dataset. Therefore, the radar echo images can well reflect the important properties of rainfall particles.

#### 2.2.2. Data Preparation

HKO-7: We followed the processing procedure for the HKO-7 similar to that described in [21]. Firstly, the Mahalanobis distance of the pixels is calculated to detect and remove outliers. Then, pixels with values less than 0 dBZ and greater than 71 dBZ were filtered out to eliminate other types of noise. A sliding window with a sequence length of 20 frames was used to slice all the radar images. We selected 9600 samples as the training set and 1600 samples as the test set. Each sample contains 20 frames of radar echo images. Each echo image frame was processed into 256 × 256 single-channel images. We used the past 10 frames of radar echoes to predict the echo images of the next 10 timestamps, that is, to predict the evolution of the echo in the next hour.

HMB: Firstly, echo images with an effective echo area of less than 20% of the whole image were removed. Then, pixels with values ranging from 0 dBZ to 80 dBZ were retained in the echo image, and the pixels were processed as outliers. We used a sliding window of length 20 to slice the echo sequence. The processed dataset was divided into a training set of 15,600 sequences and a test set of 2000, each sequence consisting of 20 frames, with the first 10 frames for input and the last 10 frames for prediction. In order to minimize the image while maintaining the resolution of the image, the original radar echo image was trimmed to remove the boundary region and retain the 768 × 768 image area in the center, and the image was further compressed to 256 × 256 by the bilinear interpolation method.

## 3. Proposed Method

### 3.1. Cascaded Dual Attention Module

In the evolution of radar echoes, the echo motion that occurs at a certain moment is affected by the echo information of multiple historical moments. Therefore, it is vital to find the spatiotemporal information for high-quality radar echo extrapolation. The information comes from the information flow composed of multiple historical moments and is most relevant to the input of the current moment. We propose a cascaded dual attention module that performs an interaction between the input feature and the memory feature. The structure of the CDAM is shown in Figure 7. CDAM consists of dual SAM connected in cascading mode and receives two inputs, the input feature $H_{t}$ at the current moment and the processed memory state feature $C_{t-1}^{\prime }$.

The SAM for $H_{t}$: The input feature $H_{t}$ passes through the first SAM, which enhances the spatial dependence between distant features in the radar echo image. The process remains the same as with the SAM described in Section 2.1.2.

The SAM for $C_{t-1}^{\prime }$: The treatment of $C_{t-1}^{\prime }$ differs from that of $H_{t}$ in that $Q_{c^{\prime }}$ is not obtained by $C_{t-1}^{\prime }$ via 1×1 convolution but is supplied directly by ${\hat{H}}_{t}$. Guided by the input feature, the SAM searches the most related feature of the radar echo movement at the current time in the historical information.

The modeling of long-term historical information is the basis of short-term prediction. The memory state feature $C_{t-1}^{\prime }$ stores the long-term information from multiple historical moments, which contains the movement trend of radar echoes and the echo features that need to be focused on. There is more instantaneous echo information present in the input features. Moreover, the connection mode inside the CDAM can effectively extend the forward propagation path of information and improve the feature extraction ability of the model.

### 3.2. Multi-Layer Attention Module

Considering the temporal characteristics of radar echo extrapolation, we find that relevant features in the past have a great influence on the prediction results at the current moment. In traditional networks used for radar echo extrapolation, the memory cell tends to focus on local spatial features and ignores long-term global dependent information.

The operation process of this module is divided into two steps, including the enhancement of historical key information and the extraction of relevant features, as shown in Figure 8. The memory information flow Clist($C_{t-1-\tau }:C_{t-1}$) is composed of multiple successive memory states in the past. Firstly, 3D-CBAM is responsible for enhancing and fusing the spatiotemporal features in the memory information flow. This process is described in detail in Section 2.1.3. The memory state acted on by 3D-CBAM highlights important spatial regions in the echo image where high-intensity echoes may occur. Then, the processed memory state feature $C_{t-1}^{\prime }$ retains the same dimension as the input feature $H_{t}$ after reshaping and 1×1 convolution. Finally, in order to fully extract the spatiotemporal information related to the current moment in the memory information flow, there is an interaction between the input feature $H_{t}$ and the memory feature $C_{t-1}^{\prime }$ occurring in the CDAM. The MLAM is formulated as follows:(14)$$C_{t-1}^{\prime }=Conv\left(reshape\left(Clist+3D-\mathrm{CBAM}\left(Clist\right)\right)\right).$$
(15)$${\hat{C}}_{t-1}=\mathrm{CDAM}(C_{t-1}^{\prime },H_{t}).$$

For radar echo extrapolation, the evolution of echoes at the current moment is affected by the echoes of multiple historical moments. In traditional ConvRNNs used for radar echo extrapolation, the memory cell tends to focus on local spatial features and ignores long-term global dependent information. Our proposed MLAM acts on multiple memory cells of the spatiotemporal sequence network, fully pays attention to and captures the key information in the memory information flow and effectively updates the memory information flow at every moment.

### 3.3. ConvRNNs with MLAM

For uniformly modeling radar echo movement trends and transient changes, a ConvRNN structure embedded with an MLAM is proposed in Figure 9. The network we built is stacked in 3 layers to extract spatial features and mitigate gradient vanishment. Except for the lowest layer, the input of each layer is the output of the adjacent lower layer, and the output of the highest layer reflects the echo situation at the next moment. The memory information flow is established in each layer to store the historical memory state information of multiple moments in the past. It is constantly updated with the time step and always records the key information. Before the memory state propagates forward to each block, it passes through the MLAM, where it interacts with the input features provided by the lower layer. Then, the memory state is updated in the block and transferred to the memory information flow in preparation for the echo extrapolationat future moments. The processing process of the network is as follows:(16)$${\hat{C}}_{t-1}^{l}=\mathrm{MLAM}\left(Clist,H_{t}^{l-1}\right).$$
(17)$$H_{t}^{l},C_{t}^{l}=\mathrm{block}\left(H_{t}^{l-1},H_{t-1}^{l},{\hat{C}}_{t-1}^{l}\right).$$

The application of the MLAM realizes the interaction of echo information across time slices, so that the network focuses on important features closely related to the prediction results, which better alleviates long-term memory loss in radar echo extrapolation.

## 4. Experiment

### 4.1. Experimental Settings

To verify that the MLAM is a universal module for improving the performance of spatiotemporal sequence networks, we selected five classical ConvRNNs commonly used for the radar extrapolation task, including ConvLSTM, ConvGRU, PredRNN, PredRNN++ and MotionRNN. The five original networks were compared with the ConvRNNs embedded with an MLAM. The implementation of the original model and MLAM+ConvRNNs is described in Section 2.1.1 and Section 3.3, respectively. In addition, using ConvLSTM as an example to evaluate the effects of MLAM submodules, ablation experiments were performed for ConvLSTM, CDAM+ConvLSTM, 3D-CBAM+ConvLSTM and MLAM+ConvLSTM.

In order to accelerate the training process, Each echo image with the shape (1, 256, 256) was subsampled to the shape (64, 64, 64) before passing through the ConvRNNs. The blocks in each layer of ConvRNNs has 128, 64, 64 convolution kernels, all of which are 3 × 3 in size. Finally, the output with the shape (64, 64, 64) of ConvRNNs was transformed to the shape (1, 256, 256) as a prediction of the current moment. The Adam optimizer is the most commonly used optimization algorithm for deep neural networks. Compared to other optimizers, it has a faster convergence rate and a more stable downtrend when training ConvRNNs [33]. All models used the Adam optimizer with a learning rate of 0.0001. The batch size was set to 4. Moreover, mean square error (MSE) was used as the loss function. All experiments is implement with Pytorch using four RTX 2080Ti GPUs.

### 4.2. Evaluation Indicators

In order to make a more reasonable analysis of the experimental results, we used image quality evaluation indicators and meteorological evaluation indicators to jointly evaluate multiple models. The structural similarity index measure (SSIM), as an image quality index, was applied to our models. The Critical Success Index (CSI) and Probability of Detection (POD) were used as meteorological indicators to evaluate the results. The threshold method was used to focus on the pixel value in the echo map. Firstly, the pixel values in the prediction echo map and the ground truth echo map were converted by thresholding. If the pixel value was greater than the given threshold, the corresponding value of the pixel point was recorded as 1, otherwise it was recorded as 0. Then, we calculated the numbers of true positive points TP (prediction = 1, truth = 1), false negative points FN (prediction = 1, truth = 0) FP (prediction = 0, truth = 1).
(18)$$\mathrm{CSI}=\frac{TP}{TP+FP+FN}.$$
(19)$$\mathrm{POD}=\frac{TP}{TP+FP}.$$

### 4.3. Comparative Study

In general, precipitation may occur when the echo intensity is generally above 20 dBZ, and if echo intensity exceeds 30 dBZ, it is considered to have strong precipitation. Values of 20 dBZ, 30 dBZ and 40 dBZ were selected as thresholds to evaluate the prediction accuracy of models. Compared with the HKO-7, the strong echo area of the HMB dataset is large and concentrated. It means that the proportion of heavy rain in the HMB dataset is much higher than that in the HKO-7 dataset, as shown in Table 1. Therefore, the scores of indicators on the HMB are generally higher than those of HKO-7.

Table 2 shows the overall evaluation results of radar echo extrapolation for 30 min and 60 min in the HKO-7 and the HMB. Compared with the standard model, the performance of CSI and POD at all thresholds of the models containing MLAMs is improved to varying degrees. It is already evident in the first 30 min of extrapolation and becomes more apparent with the lead time. We can see that MLAM+ConvLSTM has the most significant improvement in each metric on both datasets, and its prediction performance is almost identical to PredRNN and PredRNN++. This may be because, compared to other models, ConvLSTM is less capable of modeling time and space, and the MLAM promotes ConvLSTM to memorize critical information across time slices. After an hour of extrapolation, it is obvious that the higher the threshold, the greater the performance improvement of each MLAM+ConvRNN. For example, when the thresholds are 20 dBZ, 30 dBZ and 40 dBZ, the CSI of MLAM+ConvLSTM increases by 1.3%, 2% and 4.6%, and POD increases by 0.8%, 1.5% and 6.2% from the HMB, respectively, compared with ConvLSTM. This can be attributed to the fact that MLAM makes ConvRNNs pay more attention to high-intensity echoes, which helps predict areas of heavy precipitation. The results of experiments on the public dataset HKO-7 find that MLAM+MotionRNN performs best in all indicators, especially CSI30 and POD30, which reach 0.218 and 0.261, improving by 15% and 18.7% over ConvGRU. With the analysis of difference information between adjacent time slices, MotionRNN can better judge the motion trend of echoes, and the MLAM helps the model to extract important spatiotemporal information to guide this process. Analyzing the HMB dataset, MLAM+PredRNN and MLAM+PredRNN++ lead the way in more metrics. The MLAM makes the predicted echo intensity of these two models higher than the actual echo intensity, which is of great significance for heavy precipitation nowcasting, because a missed alarm of heavy precipitation is more serious than a false alarm. Moreover, considering the timeliness of the radar echo extrapolation task, we calculated the time required for all models to complete one extrapolation, which is less than 6 min. Therefore, all models are feasible under specific operational settings.

To better compare all the results, we picked an instance from each of the two datasets and visualized them. As shown in the black box in Figure 10, the radar echo pattern we selected in the HKO-7 contains a large area of a high echo. It can be seen that MLAM+PredRNN, MLAM+PredRNN++ and MLAM+MotionRNN retain the high echo region better with the extrapolation time. Among them, MLAM+MotionRNN has the best results, and its prediction of the strong echo region is large and the intensity is higher, which can greatly avoid the dangerous situation of missing heavy precipitation. In Figure 11, there is a short process of high echo aggregation and extinction in the black box regions of the selected radar echo map in the HMB. A high-intensity red echo appeared below the echo image at t = 5, and then the echo moved to the lower right with the increase in the area. At t = 17, the red echo gradually disappeared. ConvLSTM and ConvGRU failed to capture this important evolution and only made rough predictions of echo shape. MLAM+PredRNN and MLAM+PredRNN++ correctly predicted echo changes with a stronger global perspective, precisely reinforcing the processing of high echoes.

The original ConvRNNs have insufficient ability to predict echo shape, especially ConvLSTM and ConvGRU, which show that echo boundaries between different intensities become blurred, and the quality of predicted images deteriorates over time. MLAM+ConvLSTM and MLAM+ConGRU both obtain more accurate prediction results, which improve the prediction of echo details and capture the motion trend of radar echoes to achieve a better long-term prediction effect. PredRNN, PredRNN++ and MotionRNN use spatiotemporal memory cells to store and process echo information. Therefore, their predictive performance is better than ConvLSTM.

### 4.4. Ablation Experiments

Our proposed MLAM consists of 3D-CBAM and CDAM. To verify the effectiveness of each module, different types of attention schemes are applied to the ConvLSTM model, including 3D-CBAM+ConvLSTM, CDAM+ConvLSTM and MLAM+ConvLSTM. Table 3 shows the results of the four scenarios in two datasets. We observed that MLAM+ConvLSTM has the best effect, CDAM+ConvLSTM is better than 3D-CBAM+ConvLSTM in terms of overall indicators and the ConvLSTM method performs the worst. It is worth noting that experiments on the HMB dataset found that MLAM+ConvLSTM had lower POD scores on low echoes than 3D-CBAM+ConvLSTM and CDAM+ConvLSTM, which is likely due to the reduced attention to some weak echoes after using the MLAM.

To verify the effectiveness of the MLAM for high-echo processing, the CSI and POD curves at 40 dBZ at different forecast moments are described in Figure 12. At any given moment, the CSI and POD scores of MLAM+ConvLSTM are consistently higher than the original ConvLSTM. And, over time, MLAM+ConvLSTM becomes more competitive and performance decays more slowly. The scores of CDAM+ConvLSTM are generally higher than those of 3D-CBAM+ConvLSTM, which indicates that the strong echo relationship in the CDAM processing space is more predictable than that in the 3D-CBAM processing time. The MLAM can simultaneously perform feature extraction and modeling of strong echo information in time and space, so it can achieve the best results.

The visualization of the radar echo extrapolation is shown in Figure 13 and Figure 14. There is a large area of strong echoes in the black box in Figure 13, and the echo intensity in this area tends to be stable with time. The strong echoes predicted by ConvLSTM have a clear tendency to weaken with the passage of time. The 3D-CBAM+ConvLSTM, CDAM+ConvLSTM and MLAM+ConvLSTM all slow down the regression speed of echoes to different degrees. Among them, MLAM+ConvLSTM can retain more strong echo regions. In the black box of Figure 14, large-area high-intensity echoes gradually evolve into echoes of different intensities with even distribution and clear boundaries. Given the strong echo region at the center of the image, the performance of ConvLSTM to predict strong echoes weakens rapidly over time. ConvLSTM using a CDAM slows the rate of echo fading by focusing more on the key regions in the prediction process. ConvLSTM can roughly predict the approximate trajectory of the edge region where the echo intensity changes. Moreover, there are more precise boundary details in 3D-CBAM+MLAM, which indicates that 3D-CBAM correctly guides the prediction process over a wider time span. The model using the MLAM combines the above advantages to achieve higher prediction accuracy and produce higher resolution echo images.

## 5. Conclusions

In this paper, we propose a general network module called MLAM to improve the accuracy of radar echo extrapolation. The MLAM is composed of widely used 3D-CBAM and newly proposed CDAM. First, we use 3D-CBAM to enhance key memory features of multiple historical moments, which dominate the evolution trend of radar echoes and enhance the long-term tracking and attention of the network for strong echoes. Then, the CDAM captures correlations between current-moment input features and historical features, which allows the network to adapt to instantaneous changes in radar echoes. Combining the above two improvements, the MLAM establishes the long-term global dependence among multi-frame echo images in time and space. It ensures that the historical echo features can guide the echo extrapolation on the current time step. This can alleviate the problems faced by the existing ConvRNN extrapolation method, such as insufficient prediction of high-intensity echo regions and the large area decline in echo intensity in the late prediction period.

We conduct comparative experiments on two radar echo datasets, the HKO-7 provided by the Hong Kong Observatory and the HMB provided by the Hunan Meteorological Bureau. The quantitative results show that ConvRNNs with MLAMs are superior to traditional ConvRNNS in the quality of echo prediction, especially for the prediction of strong echoes. In addition, the ablation experiment is set up to fully evaluate the effectiveness of the 3D-CBAM and CDAM modules. Finally, the qualitative analysis of visualization examples shows that MLAM+ConvRNNs can better model the evolution of radar echoes than ConvRNNs, so as to achieve more accurate precipitation nowcasting.

In the future, while maintaining the stability of the model and prediction performance, we will focus on long-term radar extrapolation tasks as much as possible to achieve a longer precipitation forecast. Furthermore, we will make full use of radar echo maps at different heights to further explore precipitation prediction tasks.

## Figures and Tables

**Figure 1 sensors-23-08065-f001:**
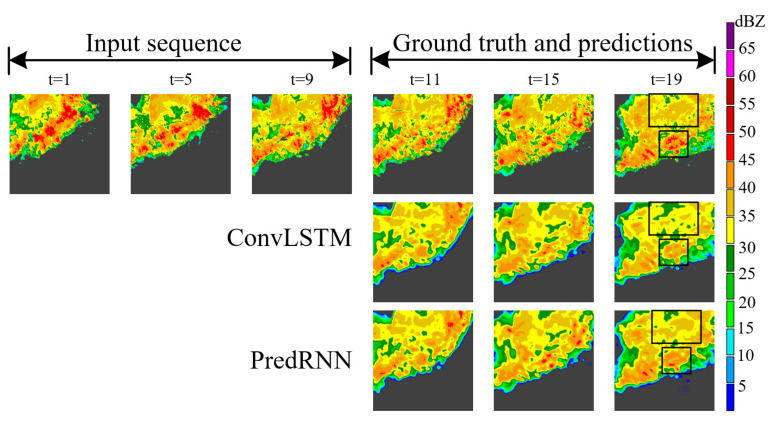
Examples of predicting the next hour of radar echo images using ConvLSTM and PredRNN from the HMB dataset. We predicted 10 frames into the future based on 10 previous frames. The black boxes indicate areas where the echo intensity declines most significantly compared to the real echo image.

**Figure 2 sensors-23-08065-f002:**
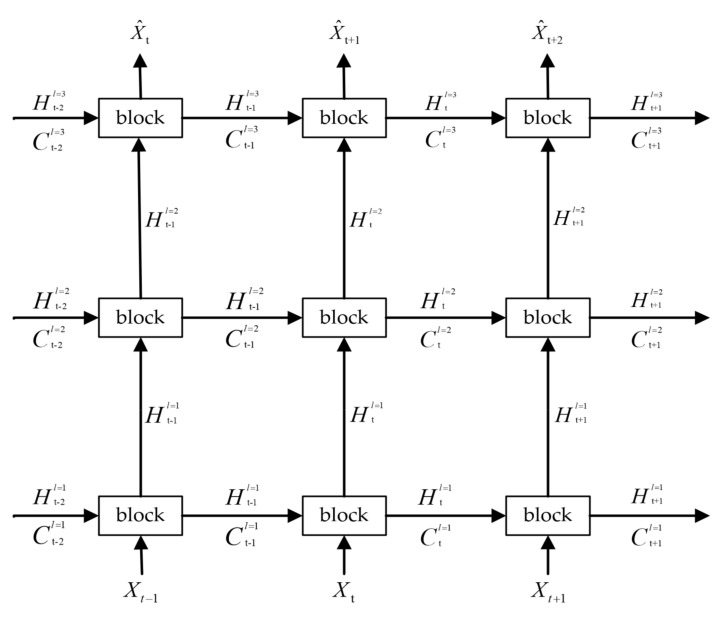
The basic model. The model consists of three layers of ConvRNNs, arrows indicate the direction of information flow and the block in the model can be replaced by ConvLSTM, ConvGRU, ST−LSTM and Casual−LSTM.

**Figure 3 sensors-23-08065-f003:**
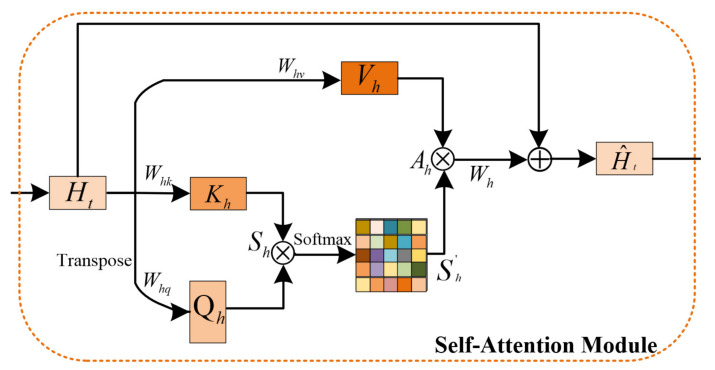
Self-attention module. $H_{t}$ is the input feature. $W_{hv}$, $W_{hk}$, $W_{hq}$ and $W_{h}$ represent the convolution operation of 1×1. $V_{h}$, $K_{h}$ and $Q_{h}$ are obtained by $H_{t}$ through convolution.

**Figure 4 sensors-23-08065-f004:**
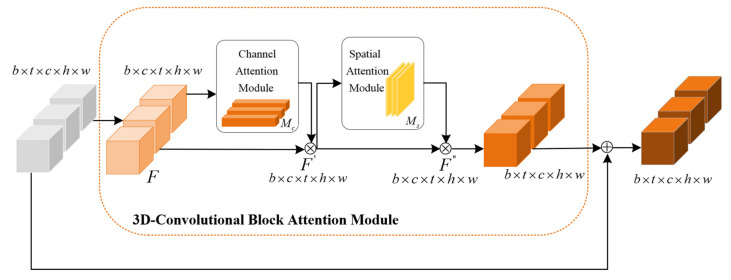
The 3D-convolutional block attention module. The input feature first passes through the channel attention module and then through the spatial attention module. The dimension of the feature vector does not change, but the shape changes to accommodate the operation between the vectors. The input feature is a five-dimensional vector, where b, t, c, h and w represent batch size, time step, number of channels, height and width, respectively.

**Figure 5 sensors-23-08065-f005:**
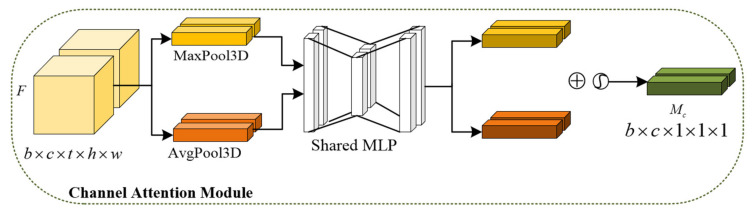
Channel attention module. b, t, c, h and w represent batch size, time step, number of channels, height and width, respectively.

**Figure 6 sensors-23-08065-f006:**
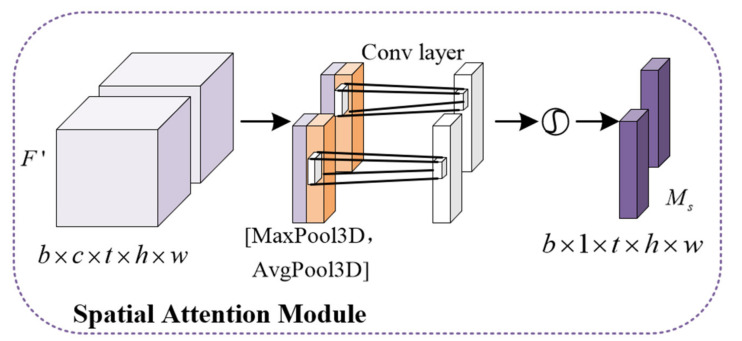
Spatial attention module.b, t, c, h and w represent batch size, time step, number of channels, height and width, respectively.

**Figure 7 sensors-23-08065-f007:**
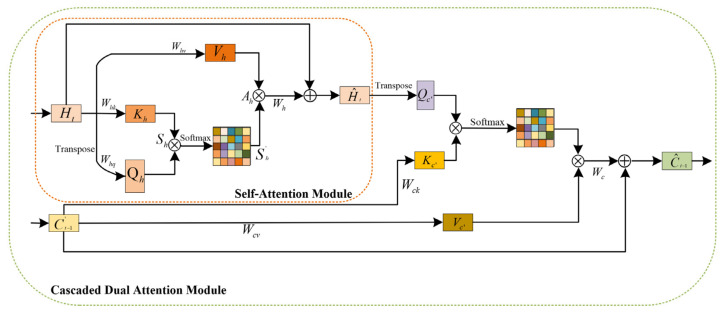
Cascaded dual attention module. $V_{h}$, $K_{h}$ and $Q_{h}$ are mappings of the input feature $H_{t}$. $V_{c^{\prime }}$ and $K_{c^{\prime }}$ are mappings of the processed memory state feature $C_{t-1}^{\prime }$. $Q_{c^{\prime }}$ in the SAM is supplied by ${\hat{H}}_{t}.$ The final output ${\hat{C}}_{t-1}$ is given by a residual connection with respect to ${\hat{H}}_{t}$.

**Figure 8 sensors-23-08065-f008:**
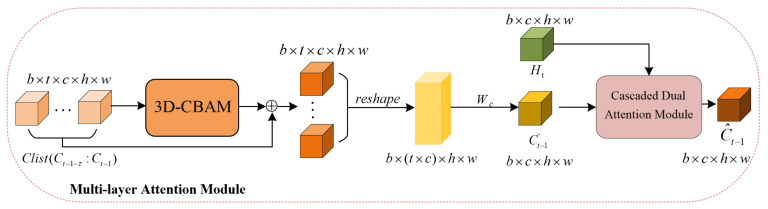
Multi-layer attention module. Clist denotes the memory information flow composed of τ moments in the past. It is a tensor with five dimensions, where the second dimension represents the length of time.

**Figure 9 sensors-23-08065-f009:**
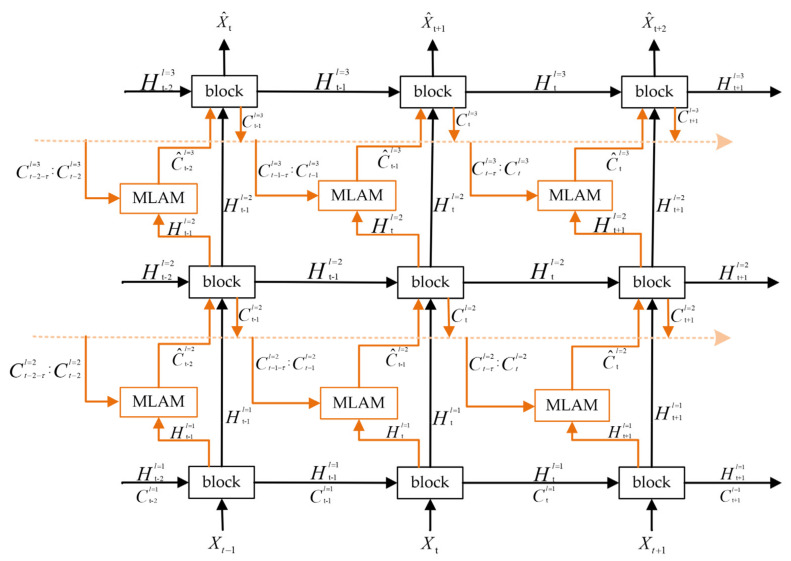
Spatiotemporal sequential networks with MLAM. The orange line denotes the information flow direction for MLAM. The orange dotted line represents the flow of memory information.

**Figure 10 sensors-23-08065-f010:**
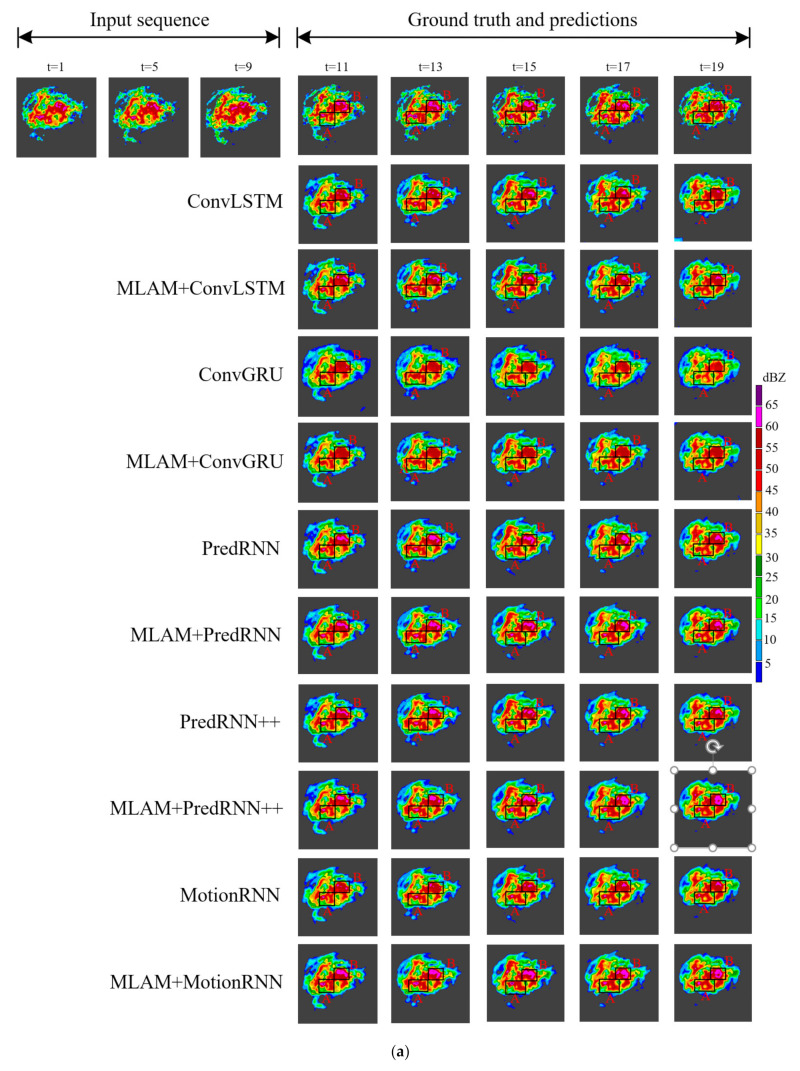
(**a**) Visualization example from HKO-7. The echo evolution occurred on 14 April 2013. Each model predicts the next 10 frames by observing 10 frames. A large area of strong echo exists in black rectangular boxes. Models with MLAM can generate better predictions. (**b**) Comparison of echo details after enlarging the black box area in (**a**). A and B are the two black rectangular regions in (**a**).

**Figure 11 sensors-23-08065-f011:**
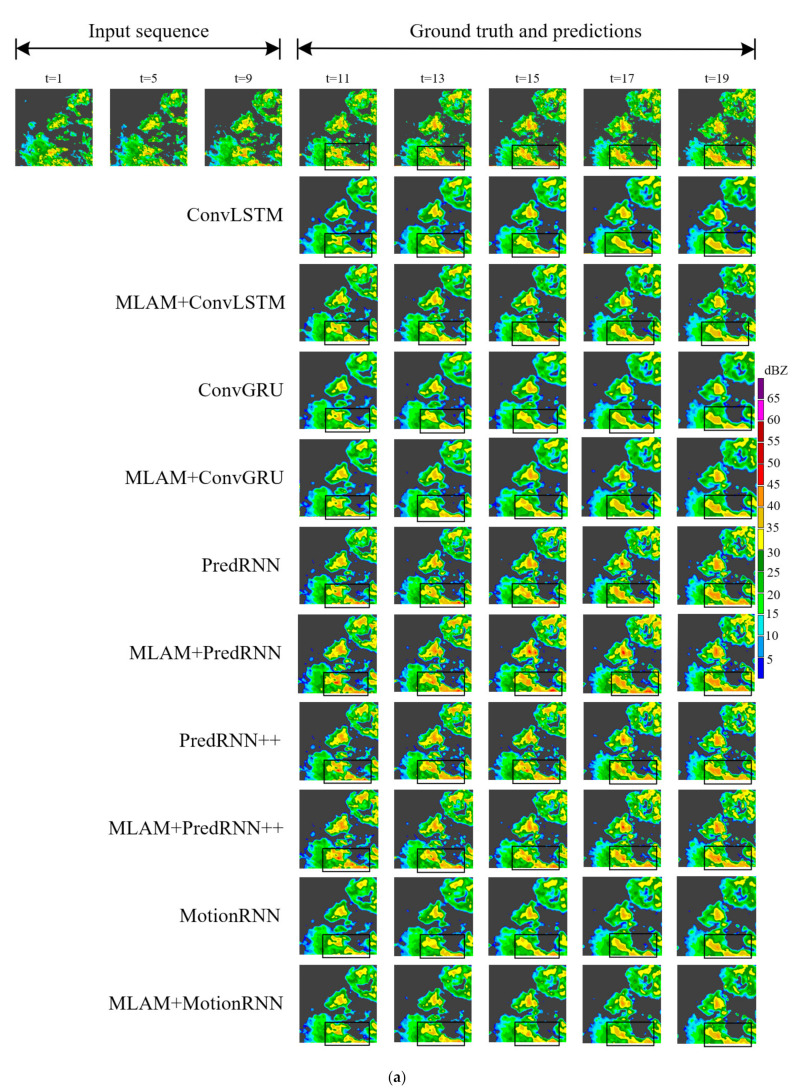
(**a**) Visualization example from the HMB dataset. The red echoes in the black box area gather to the lower right. (**b**) Comparison of echo details after enlarging the black box area in (**a**).

**Figure 12 sensors-23-08065-f012:**
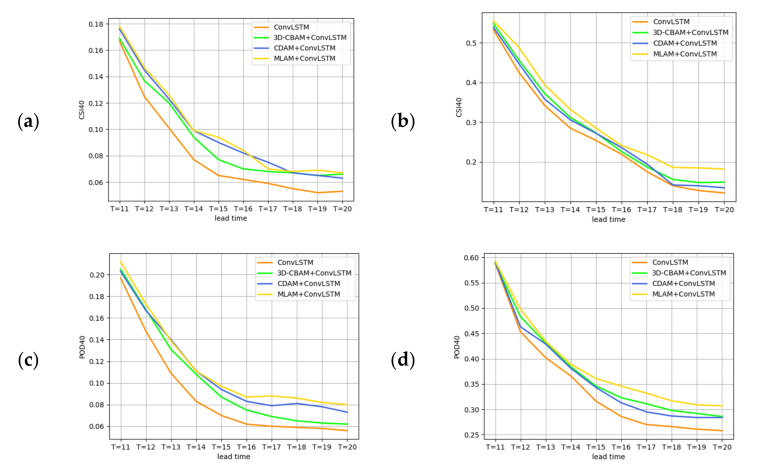
(**a**) CSI curves along different lead times at 40 dBZ threshold in the next 1 h from the HKO-7. (**b**) CSI curves along different lead times at 40 dBZ threshold in the next 1 h from the HMB. (**c**) POD curves along different lead times at 40 dBZ threshold in the next 1 h from the HKO-7. (**d**) POD curves along different lead times at 40 dBZ threshold in the next 1 h from the HMB.

**Figure 13 sensors-23-08065-f013:**
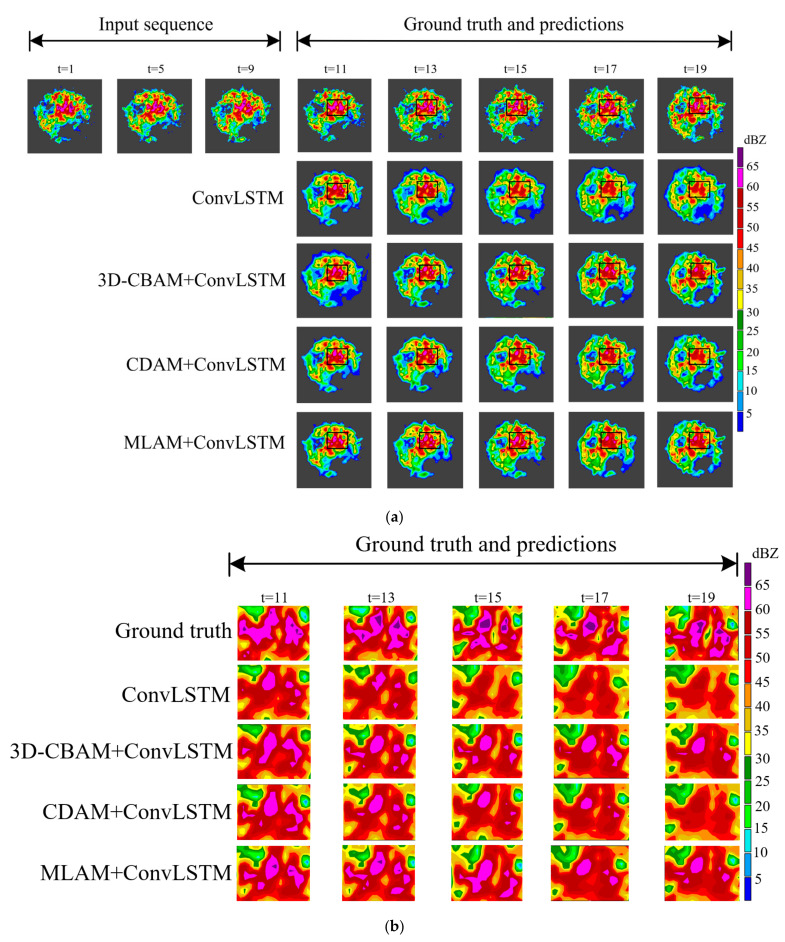
(**a**) Prediction example of ablation experiments from HKO-7 using ConvLSTM as the baseline model. Frames are displayed at a two-frame interval. This example appeared on 17 October 2012. There is a large area of strong echo in the black box area and the echo area and intensity tends to be stable with time. (**b**) Comparison of echo details after enlarging the black box area in (**a**).

**Figure 14 sensors-23-08065-f014:**
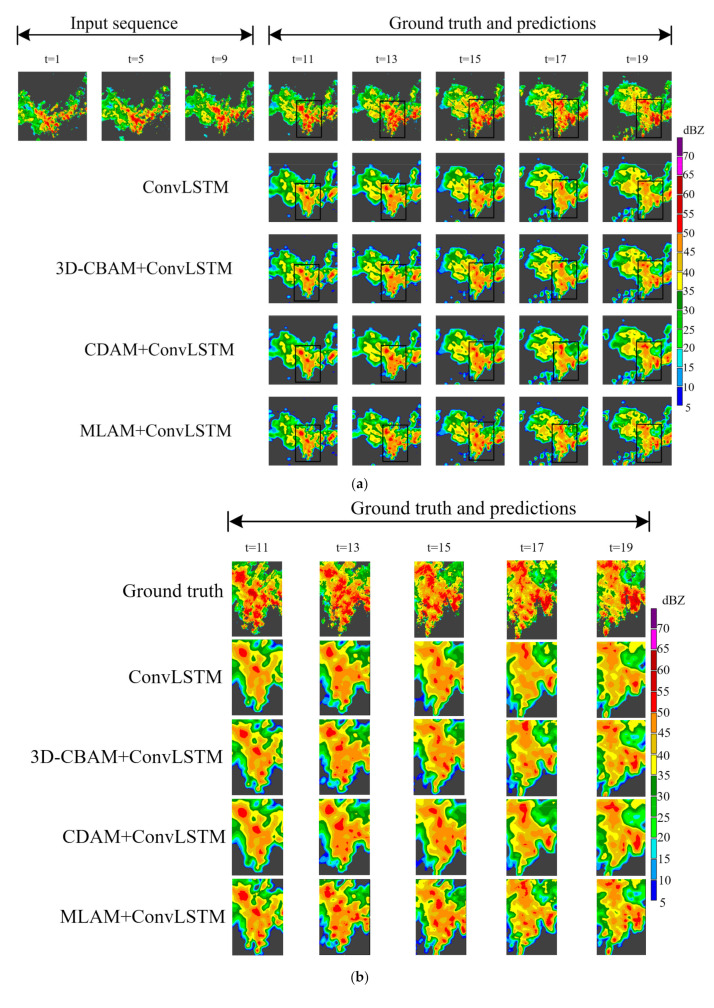
(**a**) Prediction examples of ablation experiments from the HMB dataset using ConvLSTM as the baseline model. In the black box area, the echo intensity is evenly distributed and there are clear boundaries between echoes of different intensities. (**b**) Comparison of echo details after enlarging the black box area in (**a**).

**Table 1 sensors-23-08065-t001:** Rain rate statistics in HKO-7 and HMB. When the rain rate is greater than 10 mm/h, rainfall level is considered as heavy rain.

Dataset	Rain Rate (mm/h)			
	0≤x<5	5≤x<10	10≤x<30	30≤x
HKO-7	97.09%	1.35%	1.14%	0.42%
HMB	66.19%	20.27%	9.56%	3.96%

**Table 2 sensors-23-08065-t002:** Prediction results of comparative experiments on two radar echo datasets. All models predict the next 10 frames by 10 known frames. CSI, POD and SSIM are adopted to indicate the accuracy of the prediction, where (a) is the prediction result of the next half an hour and (b) is the prediction result of the next one hour.

Dataset	Model	CSI ↑			POD ↑			SSIM ↑
		20	30	40	20	30	40	
(a)								
	ConvLSTM [9]	0.260	0.173	0.089	0.335	0.207	0.096	0.772
	ConvGRU	0.203	0.090	0.019	0.289	0.098	0.029	0.745
HKO-7	PredRNN [18]	0.317	0.191	0.097	0.405	0.214	0.123	0.802
	PredRNN++ [19]	0.323	0.202	0.108	0.418	0.229	0.127	0.822
	MotionRNN [24]	0.351	0.242	0.143	0.463	0.277	0.154	0.843
	MLAM+ConvLSTM	0.298	0.194	0.102	0.386	0.254	0.121	0.777
	MLAM+ConvGRU	0.232	0.106	0.056	0.342	0.134	0.065	0.749
	MLAM+PredRNN	0.325	0.203	0.129	0.422	0.241	0.148	0.813
	MLAM+PredRNN++	0.336	0.223	0.127	0.447	0.257	0.138	0.825
	MLAM+MotionRNN	**0.357**	**0.251**	**0.153**	**0.476**	**0.290**	**0.166**	**0.844**
	ConvLSTM	0.755	0.654	0.376	0.845	0.764	0.434	0.797
	ConvGRU	0.764	0.649	0.371	0.839	0.754	0.422	0.793
	PredRNN	0.771	0.663	0.354	0.884	0.777	0.486	0.868
HMB	PredRNN++	0.794	0.671	0.397	0.894	0.792	**0.553**	0.880
	MotionRNN	0.764	0.651	0.388	0.874	0.783	0.464	0.875
	MLAM+ConvLSTM	0.778	0.668	0.411	0.863	0.789	0.467	0.811
	MLAM+ConvGRU	0.773	0.657	0.401	0.851	0.786	0.503	0.867
	MLAM+PredRNN	0.784	0.675	0.367	0.868	**0.823**	0.543	0.871
	MLAM+PredRNN++	**0.801**	**0.678**	0.406	**0.912**	0.809	0.525	0.882
	MLAM+MotionRNN	0.787	0.656	**0.432**	0.897	0.794	0.502	**0.893**
(b)								
	ConvLSTM	0.222	0.140	0.082	0.299	0.165	0.090	0.739
	ConvGRU	0.171	0.068	0.015	0.240	0.074	0.025	0.683
HKO-7	PredRNN	0.255	0.167	0.096	0.350	0.197	0.106	0.740
	PredRNN++	0.263	0.171	0.096	0.360	0.202	0.106	0.748
	MotionRNN	0.301	0.217	0.130	0.420	0.256	0.143	0.746
	MLAM+ConvLSTM	0.248	0.169	0.101	0.355	0.211	0.116	0.748
	MLAM+ConvGRU	0.173	0.078	0.017	0.254	0.086	0.027	0.698
	MLAM+PredRNN	0.283	0.192	0.116	0.385	0.226	0.138	0.754
	MLAM+PredRNN++	0.287	0.197	0.115	0.394	0.232	0.126	**0.760**
	MLAM+MotionRNN	**0.305**	**0.218**	**0.136**	**0.422**	**0.261**	**0.152**	0.758
	ConvLSTM	0.662	0.527	0.262	0.740	0.633	0.327	0.767
	ConvGRU	0.660	0.521	0.252	0.735	0.621	0.314	0.766
	PredRNN	0.670	0.522	0.258	0.757	0.656	0.403	0.787
HMB	PredRNN++	0.686	0.544	0.292	0.770	0.678	**0.448**	0.792
	MotionRNN	0.666	0.536	0.285	0.748	0.645	0.360	0.829
	MLAM+ConvLSTM	0.675	0.547	0.308	0.748	0.648	0.389	0.784
	MLAM+ConvGRU	0.668	0.539	0.314	0.749	0.670	0.444	0.778
	MLAM+PredRNN	0.672	0.512	0.230	0.745	**0.697**	0.432	0.795
	MLAM+PredRNN++	**0.687**	0.541	0.275	**0.781**	0.651	0.396	0.803
	MLAM+MotionRNN	0.684	**0.562**	**0.329**	0.761	0.666	0.415	**0.841**

The best results are bolded. **↑** means that higher is better.

**Table 3 sensors-23-08065-t003:** Ablation study on radar echo datasets. We used CSI, POD and SSIM to measure the prediction quality in the next 1 h. ConvLSTM is the baseline and three variants were evaluated, including ConvLSTM with 3D-CBAM, ConvLSTM with CDAM and ConvLSTM with MLAM.

Dataset	Model	CSI ↑			POD ↑			SSIM ↑
		20	30	40	20	30	40	
HKO-7	ConvLSTM	0.222	0.140	0.082	0.299	0.165	0.090	0.739
	3D-CBAM+ConvLSTM	0.236	0.159	0.093	0.339	0.189	0.103	0.743
	CDAM+ConvLSTM	0.246	0.165	0.099	0.350	0.206	0.111	0.743
	MLAM+ConvLSTM	**0.248**	**0.169**	**0.101**	**0.355**	**0.211**	**0.116**	**0.748**
HMB	ConvLSTM	0.662	0.527	0.527	0.740	0.633	0.213	0.767
	3D-CBAM+ConvLSTM	0.668	0.531	0.531	**0.772**	**0.672**	**0.237**	0.780
	CDAM+ConvLSTM	0.671	0.541	0.541	0.750	0.648	0.212	0.777
	MLAM+ConvLSTM	**0.675**	**0.547**	**0.547**	0.748	0.648	0.203	**0.784**

The best results are bolded. **↑** means that higher is better.

## Data Availability

The HMB dataset in the manuscript was collected by the local meteorological department (Hunan Meteorological Bureau). The authors collaborated with the institution for scientific research, and they grant permission for the dataset to be used for research purposes. Due to laws and regulations, the disclosure of government department data is prohibited.

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
