# Peer review of "MLAM: Multi-Layer Attention Module for Radar Extrapolation Based on Spatiotemporal Sequence Neural Network"

_sensors, 2023, doi:10.3390/s23198065_

Round 1

Reviewer 1 Report (Previous Reviewer 3)

The authors have improved the paper considerably from its previous version, and have adequately addressed all the issues I had. They have clarified the data sources and treatment of the data (in terms of training sets and test data). They have also improved the presentation of the results by making it easier to understand the progression of the statistical metrics with forecast lead time.

The English is fine. There are some minor grammatical errors but nothing that interferes with understanding the work.

Author Response

Reviewer 2 Report (Previous Reviewer 2)

Dear Authors,

Thank you for re-submitting your research regarding nowcasting precipitation using radar data and neural network methods. This topic is interesting because of the necessity of improving the ability to forecast and diagnose potentially dangerous rainfall structures with a higher lead time. This fact is still unresolved, and there are many damages and fatalities each year around the World. However, there is a lot of recent research in the same direction that the one proposed in this manuscript, and this fact makes it difficult to highlight the capabilities of your proposal in front of the rest. I want to thank the Authors for the clarity of the methodology, which is one of the issues in most of the similar papers. In this way, my major concerns continue in the same line as the previous version:  

- The Authors need to explain better the data source. The description of the two datasets is vague, without any relevant information regarding the origin of the data, the limitations, etc. Have you applied any quality control to the data?

- Besides, why do you have selected the two datasets? Which are the criteria for the selection of the data?

- It is not clear at all which criteria for the predicted value=1 and true value=1. Have you considered the possibility of anomalies in the data?

- Figures 10 and 11 are very difficult to read and do not provide any relevant information. The Authors should focus on the elements that they try to highlight.

- Same with Figs 13 and 14

- The Authors should explain better the quality of a POD < 0.25 and the differences between both datasets. This point is crucial.

- Conclusions: this section needs to be enlarged. Furthermore, I miss some discussions compared with other previous research.

Best regards

Author Response

Reviewer 3 Report (New Reviewer)

Review of MLAM:Multi-layer Attention Module for Radar Extrapolation Based on Spatiotemporal Sequence Neural Network by Wang etc.

This paper tries to improve precipitation nowcasting using radar echo extrapolation through spatiotemporal sequence neural network. The paper proposed an embedded MLAM to address the short-coming of the standard ConvRNNs. The paper describes the MLAM enhancement first and then presents its prediction results by comparing with the other commonly used neural network methods. In general, this paper is well organized and its English writing is also good. It can be published with the journal.

I have some general comments for the authors to consider. First, the description about the neural network background (Section 2) and about the proposed method (Section 3) are pretty hard to follow and even hard to understand well. If the authors can improve the descriptions using more plain language sentences, the value of this paper will be improved significantly. Second, the paper uses many abbreviations that appear suddenly. Some of them do not have the full definitions or contexts, for examples, LSTM, CNN, RNN, U-Net, GAN, DGMR etc in the introduction. Some of them are used without explicit denotations, for examples, SAM, CBAM, MLP, CAM etc. in Section 2 and readers have to guess their meanings.

Minor issue:

  1. Most of the Figure captions are too simple. Some of the figures even use the same caption, for example, Figure 3 and Figure 6.
  2. Figure 1 highlights two red rectangles. It had better denote the same boxes in other time series as well. Furthermore, the lower box denotes a region where reflectivity enhances but not decays and it conflicts with its caption?
  3. Line 291: “6 min”
  4. Line 337: May need a few sentences to describe “Adam optimizer”.
  5. Incomplete sentence with caption of Figure 12.

Round 2

Reviewer 2 Report (Previous Reviewer 2)

Dear Authors,

Thank you for your answers.

I have stil some comments:

1. it is highly recommended to indicate in the text the changes (using a different color, for instance)

2. The reasons for the selection of the two datasets should be included in the manuscript

3. Please, could you include some information about radar type? (e.g. C, X, or S-band, dual or single pol, if it is a single radar o a composite)

4. please, add the information of the point 3 in the manuscript

5. Please, include the zoom results of the figures 10-14 in some way, to make more readable the information

6. I can't find the table in the new version of the manuscript. Please, include it and all the adding information

7. The conclusions have not been enlarged at all

Best regards

Round 3

Reviewer 2 Report (Previous Reviewer 2)

Dear Authors,

Thank you for answering my questions

Best regards

This manuscript is a resubmission of an earlier submission. The following is a list of the peer review reports and author responses from that submission.

Round 1

Reviewer 1 Report

This paper proposes a module that uses multiple attention mechanisms to improve the performance of spatiotemporal sequence networks to solve radar echo extrapolation. The innovation point of the article is clear and effective experiments are carried out. This paper can be accepted after minor revisions. The followings are the suggestions:

1.     In the section 2, more introduction of content related to the innovation points should be added.

2.     In the section 3, the proposed CDAM and MLAM should be treated as two equally important modules. Therefore, when introducing them, they should be presented on an equal footing instead of treating CDAM as a subheading of MLAM. The order of introduction of each module needs to be adjusted. The overall network structure can be introduced first, followed by the introduction of each component module.

3.     Necessary analysis is lacking for the data in Tables 1 and 3, such as in Table 1, it would be better to conduct some analysis on the reasons for the poor performance of the two metrics, POD20 and POD30, for MLAM+ConvLSTM.

4.     In formula (3), it is not clear how C units at multiple time steps are specifically calculated in 3D-CBAM, which is also not reflected in Figure 2. An explanation of this step is needed in this section.

5.     What is the RNN labeled in Figure 1? The naming of RNN here is inappropriate, and it is recommended to change it to "unit", "cell" or "block".

6.     For CDAM, the specific internal calculation process is not explained. And in Figure 6, it is recommended to label the two cascaded parts.

Reviewer 2 Report

Dear Authors,

After reading carefully your manuscript and comparing with other similar research based on the ConvLSM or adaptations, I need more evidences of the improvement of your technique. Because of this, you need to present a better explanation of the experiments, which should include a large list of events (that should be briefly introduced), and make a better validation.

Best regards

Reviewer 3 Report

Firstly, I would like to say that I do not possess the expertise necessary to assess the entirety of the paper. In particular, I am not familiar with the details of the different neural network configurations, and so cannot assess the strength or originality of the model/module configurations.

On the other hand I believe that there are aspects of the radar extrapolation science and verification presented in this paper that need attention.

The references to existing radar extrapolation techniques seems inadequate to me. The authors rightly mention TREC and SCIT, but these are dated systems. More sophisticated schemes such as STEPS e.g., Seed et al, 2013) and LINDA (Pulkkinen et al. 2018) which have become the benchmarks in this field and are used operationally. It may also be worth mentioning the work of the DeepMind team (Ravuri et al. 2021). These are the extrapolation methods that more realistically represent the state of the science.

The authors compare their methods internally to identify which performs best, but do not compare their results to existing methods. It is not essential for this paper, but it would be interesting to see how the method performs compared to a baseline simple extrapolation scheme.

Verifying the value of forecasts such as these is problematic and the statistics provided here are inadequate in capturing the full extent of the error characteristics and thereby informing the strengths and weaknesses of the different models. Having said that, I believe that they give enough guidance to understand which version is working best. The difficulty I have with the statistics presented is that they appear to be global, in that they are calculated and aggregated for all lead times combined. Whether this is or isn't the case is not clear as the authors don't say whether this is for a particular lead time or all lead times combined. Usually in this type of paper I expect to see how the assessment varies with lead time, so that I can assess the time for which a "good" forecast is provided and whether the forecast performance is behaving as expected.

The value of radar echo extrapolation lies in its ability to provide meteorologists and those who rely on them with forecasts of storms and precipitation in a timely manner whereby responses can be made in very short periods. Therefore, one of the critical aspects of these systems is the speed with which forecasts can be produced and delivered. With a complex computational method such as the one presented in this work, it is important to know whether it would be practical to implement this methodology in an operational setting, considering the time taken to process real-time radar data and then run the forecasts every 6 minutes.

If the authors are less interested in producing a working radar extrapolation method that can be implemented in real time, and are more interested in investigating the use of the MLAM for this type of application, then I believe this paper is fine. But the authors need to make that clear. If they are attempting to create a genuine nowcast system, then they need to increase the level of detail of the verification (e.g., lead time dependence, and a possible comparison to a simple extrapolation model), and include a discussion of how the computational efficiency and if there are situations where the model has weaknesses.

Round 2

Reviewer 2 Report

Dear Authors,

You have improved some minor aspects of your manuscripts, but the main issues (see my previous report) have not been solved.

In this current state, the research can not been accepted.

Best regards